# Charge Diffusion and Repulsion in Semiconductor Detectors

**DOI:** 10.3390/s24227123

**Published:** 2024-11-06

**Authors:** Manuel Ballester, Jaromir Kaspar, Francesc Massanés, Alexander Hans Vija, Aggelos K. Katsaggelos

**Affiliations:** 1Department of Computer Sciences, Northwestern University, Evanston, IL 60208, USA; manuelballestermatito2021@u.northwestern.edu; 2Siemens Medical Solutions USA Inc., Hoffman Estates, IL 60192, USA; jaromir.kaspar@siemens-healthineers.com (J.K.); francescdassis.massanesbasi@siemens-healthineers.com (F.M.); hans.vija@siemens-healthineers.com (A.H.V.); 3Department of Electrical and Computer Engineering, Northwestern University, Evanston, IL 60208, USA

**Keywords:** semiconductor detectors, high-energy radiation detection, charge dynamic modeling, charge diffusion, coulomb repulsion, charge cloud distribution, 3D Gaussian expansion

## Abstract

Semiconductor detectors for high-energy sensing (X/γ-rays) play a critical role in fields such as astronomy, particle physics, spectroscopy, medical imaging, and homeland security. The increasing need for precise detector characterization highlights the importance of developing advanced digital twins, which help optimize the design and performance of imaging systems. Current simulation frameworks primarily focus on modeling electron–hole pair dynamics within the semiconductor bulk after the photon absorption, leading to the current signals at the nearby electrodes. However, most simulations neglect charge diffusion and Coulomb repulsion, which spatially expand the charge cloud during propagation due to the high complexity they add to the physical models. Although these effects are relatively weak, their inclusion is essential for achieving a high-fidelity replication of real detector behavior. There are some existing methods that successfully incorporate these two phenomena with minimal computational cost, including those developed by Gatti in 1987 and by Benoit and Hamel in 2009. The present work evaluates these two approaches and proposes a novel Monte Carlo technique that offers higher accuracy in exchange for increased computational time. Our new method enables more realistic performance predictions while remaining within practical computational limits.

## 1. Introduction

Single-photon semiconductor detectors are widely used in fields such as astronomy, particle physics, spectroscopy, radiology, and homeland security. These detectors are highly effective in precisely identifying both the 3D location of photon–detector interactions and the deposited energy of incoming high-energy photons. These detectors use thick crystals to stop most of the incoming radiation, and they are often made of materials such as Ge, Si, CdTe or CdZnTe as well as emerging materials like Hg$I_{2}$, Pb$I_{2}$ and TlBr [1,2,3,4,5,6].

In recent years, there has been significant progress in the development of accurate digital twins that closely replicate the functioning of such detectors [7,8,9,10,11,12,13,14,15,16,17,18,19,20]. These simulators enhance our ability to reconstruct the events (3D position and energy) more efficiently. For instance, some of these computational models incorporate spatially dependent crystal properties and defects, providing refined event reconstruction for customized detector configurations and compensating for crystal impurities [21,22,23,24,25,26].

A basic design for these detectors is the planar configuration, featuring a cathode on one side and an anode on the opposite side, as seen in Figure 1a. Nevertheless, it is important to emphasize that our presented model is not influenced by a particular detector type and could be applied to more advanced configurations, such as the common pixelated geometry (that takes advantage of the small pixel effect on the anode side) [27] or Frisch grid structures [28].

Between the detector electrodes, a bias electric field $E_{b}$ is established (see Figure 1a). When a photon is absorbed, it undergoes an initial conversion process, such as the photoelectric or Compton effect, which then generates *P* electron–hole pairs, typically ranging from tens to hundreds of thousands, depending on the deposited energy and the ionization energy of the crystal. These free electrons and holes are then driven toward their corresponding electrodes by the bias E-field: the holes migrate toward the cathode along the field lines, while the electrons are pushed toward the anode. The digital twins must accurately replicate how these electrons and holes move as they are drifted through the detector’s bulk. Understanding the temporal and spatial propagation of the charges enables the accurate reconstruction of the current signals at the electrodes. In particular, the so-called Shockley–Ramo theorem explains how to emulate the detector signals knowing the movement of the free electron and holes [29,30,31,32,33].

Figure 1b displays the simulated current signals at the anode location produced with a modern simulator [20]. For this visual example, we consider a representative planar CZT detector with the reasonable dimensions of 1×1×1 c$m^{3}$. Note that the photon interaction event here occurred right at the center of the cube, at the coordinates $\left(x_{0},y_{0},z_{0}\right)=\left(0.5,0.5,0.5\right)$ cm of the detector, and the presumed deposited energy was 140.5 keV, corresponding to the gamma photons typically used in medical SPECT applications and emitted by a source of Technetium-99m. To simplify the model, we assume a strong bias E-field, approximated as a uniform constant field of $E_{b}=1000$ V/cm throughout the crystal, which was created by the applied linear voltage induced at the electrodes (see Figure 1a) [34,35,36]. While this planar detector configuration and constant-field assumption serve as a practical scenario for demonstration, they are not limitations of our method for analyzing diffusion and repulsion processes, which can be extended to more complex detector geometries [37] and variable E-fields [38,39,40].

In this example, the free electrons generated within the CZT crystal reach the anode more quickly than the holes reach the cathode. This is due to the significantly higher mobility of electrons, which is approximately 1000 c$m^{2}$/Vs, compared to the mobility of holes, which ranges between 10 and 100 c$m^{2}$/Vs [41]. Note that the amount of charges fluctuates with time and decreases as a result of the trap-induced recombination [42,43], leading to a slight decay in the signals over time (see Figure 1b). One can additionally see a drop in the electron signal right after the electron collection at 0.5 μs. The electron drift velocity can be estimated as the product of the charge mobility with the bias E-field, $v=\left(10^{3}{\mathrm{cm}}^{2}/\mathrm{Vs}\right)\left(10^{3}V/\mathrm{cm}\right)=10^{6}$ cm/s in the simulated scenario plot in Figure 1b. This implies that electrons travel from the midpoint of the detector (at z=0.5 cm) to the anode (at z=0 cm) in 0.5 μs, as seen in Figure 1a,b. Observe in Figure 1b that the electrons contribute more than the holes to the total anode signal, which happens because of their higher drift velocity and their increasing proximity to the anode over time. It must be emphasized that another similar signal, although with a negative sign, could be read at the cathode location (not shown in the figure).

However, many of the simulators from the literature [7,8,14,19,20,25,44] fully overlook the effects of charge diffusion and Coulomb repulsion. They rather consider the cloud as a point charge that moves following the one-dimensional paths of the bias E-field. Other simulators omit only the self-repulsion effect [13,16], as this effect is particularly demanding from the computational point of view [45]. While we should acknowledge that diffusion and repulsion are second-order effects that often have a relatively low impact on the current signal, they are responsible for spreading the charge cloud in three-dimensional space as it traverses the crystal, as shown in Figure 1c.

Many simulators exclude these two effects not because they are insignificant but due to the increased model complexity and computational time associated with their inclusion. In many applications, neglecting them still yields acceptable approximations. However, as new-generation detectors demand higher energy and spatial resolution, these subtle effects are becoming increasingly important in refined simulations [18]. Quantifying the specific contribution of diffusion and repulsion is challenging, as their impact strongly depends on numerous factors, including the detector geometry, the semiconductor material, the energy of incident photons, the location of the interaction, and the operation conditions (such as temperature). In any case, to highlight the importance that these two effects have in real scenarios, consider an event occurring near or over the interpixel gap between two anodes. In this situation, a basic model that treats the charge injection as a delta distribution could lead to complete current leakage or their collection at only one anode pixel. In contrast, accounting for the expansion of the charge cloud (which can grow to hundreds of microns) could result in the realistic phenomenon of shared collection of the electron cloud at the adjacent anodes, significantly changing the signals shape.

Our present study analyzes these second-order effects in depth. First, we will evaluate two prevalent models previously employed to integrate these effects effectively: (i) the analytical study developed by Gatti et al. [46] in 1987 and (ii) the stochastic method devised by Benoit and Hamel [47] in 2009. Gatti’s model treated diffusion and Coulomb repulsion as separate phenomena, each modeled by a distinct partial differential equation. This work has influenced many subsequent relevant analyses [48,49,50,51,52].

Built upon Gatti’s results, the more recently developed Benoit–Hamel (BH) method consolidates both the diffusion and repulsion effects together into a single mathematical model. This approach assumes that the charge distribution maintains a simple three-dimensional Gaussian shape over time with a mean located at the center of the cloud and an evolving standard deviation that slowly increases. Our proposed approach further extends the BH model and proposes that the charges spread out following a generalized normal distribution (GND). This distribution offers greater flexibility than the simple Gaussian model and allows us to accurately match the shape of the real intricate charge spread.

## 2. Theoretical Background

For the moment, we will focus solely on the diffusion and repulsion of charges, neglecting (i) the charge drift due to the external E-field and (ii) the charge fluctuation due to additional pair generations, recombination, or trapping-detrapping processes. Observe that it is possible to adopt a local coordinate system centered on the electron cloud, which moves at the electron drift velocity [48]. This choice of reference frame can simplify the analysis by considering the electron cloud as stationary relative to its surrounding environment (see Figure 1c,d). The core of our analysis then becomes the well-known diffusion-repulsion equation, which is formulated as
(1)$$\partial_{t}n=D\nabla^{2}n-\nabla \cdot \left(\mu nE\right)$$
Here, μ represents the mobility of the charge carriers, *D* denotes the Einstein diffusion coefficient, and E is the field generated by the cloud itself. Importantly, *D* is not an independent parameter, as it can be derived from the semiconductor temperature *T* and the carrier mobility. Following the Einstein–Smoluchowski relation [53], $D=\mu k_{B}T/q$, where $k_{B}$ is the Boltzmann constant and *q* is the elementary charge. Equation (Equation 1) thus describes the time- and space-dependent evolution of the electron concentration n(x,y,z,t). A similar equation and approach are applicable to holes, considering their corresponding mobility and diffusion coefficient. According to Gauss’s law, the repulsive Coulomb field generated by the charges within the cloud is expressed as
(2)$$\nabla \cdot E=\frac{\rho }{\epsilon }$$
Here, ρ=q n is the electron density (C/c$m^{3}$) and ϵ is the permittivity of the crystal. We then aim to solve a system with two coupled partial differential equations (PDEs) with Equations (Equation 1) and (Equation 2). The self-repulsion field E of the cloud should not be confused with the externally applied bias electric field $E_{b}$ between the electrodes, as shown in Figure 1a.

Both the diffusion process and Coulomb repulsion inherently lack directional preference. When the *P*-generated electron–hole pairs are located deep within the central bulk of the detector, they are minimally affected by boundary effects, and it is expected that they will maintain radial uniformity [46]. The characteristic broadening of the distribution is then expected as the concentration evolves in time (see Figure 1c,d). We can now rewrite the equations and variables using spherical coordinates (r,θ,ϕ). If we multiply both sides of Equation (Equation 1) by *q* and perform the 3D integral over a ball with radius *R*, denoted by the open set $\Omega =B\left(0,R\right)\subset R^{3}$, we obtain the following result:(3)$$\int_{\Omega }q\partial_{t}ndV=\int_{\Omega }q\left(\frac{D}{r^{2}}\partial_{r}\left(r^{2}\partial_{r}n\right)-\mu \nabla \cdot \left(nE\right)\right)dV$$
The left-hand
(4)$$\int_{\Omega }q\left(\frac{D}{r^{2}}\partial_{r}\left(r^{2}\partial_{r}n\right)\right)dV=D\left(\partial_{RR}Q-\frac{2\partial_{R}Q}{R}\right)$$
The repulsion part on the right-hand side of Equation (Equation 3) can be simplified using the Gauss divergence theorem [54],
(5)$$\int_{\Omega }q\left(\mu \nabla \cdot (nE)\right)dV=\mu \frac{\partial_{R}Q}{4\pi R^{2}}\frac{Q}{\epsilon }$$
Note that the Coulomb field E was calculated, as a function of the total charge *Q*, by solving analytically the Gauss law [55] over the conveniently selected ball Ω. We have finally derived a simplified reformulation of the initial coupled partial differential equations from Equations (Equation 1) and (Equation 2) as
(6)$$\partial_{t}Q=D\left(\partial_{RR}Q-\frac{2\partial_{R}Q}{R}\right)-\mu \frac{\partial_{R}Q}{4\pi R^{2}}\frac{Q}{\epsilon }$$
Equation (Equation 6) does not have a closed-form analytical solution given a general initial condition Q(R,t=0). However, we can solve it using numerical tools. Despite its apparent simplicity, special care should be taken due to the nonlinear behavior. We have employed a customized implicit finite difference method (IFDM) [56] and solved the resulting nonlinear discrete equations (at each time step) with the Newton–Raphson method. More details are provided in Appendix A. We analyze the propagation of charges in a time range of (0,1] μs, with T=200 time steps, and a radial space interval of (0,400] μm, where N=400 represents the spatial steps.

While the current numerical approach successfully provides the desired solution, it presents two significant challenges. First, it is important to note that standard computers require approximately 3 to 7 s to compute this solution. Although this might not appear excessive at first glance, efficient simulators are typically designed to operate within a timeframe of tens of milliseconds, as they often must be evaluated thousands of times to address inverse problems. Second, these digital twins do not only account for diffusion and repulsion but also for the charge drift due to the external field and the charge fluctuations, and incorporating the numerical solutions for diffusion–repulsion into a generally complicated simulator framework is not straightforward.

As a result, research efforts focus on developing approximate solutions that are computationally efficient and can be easily integrated into broader simulation frameworks. In this work, we will utilize the precise numerical results from the IFDM method solely for comparison purposes, allowing us to evaluate the accuracy of two commonly used efficient approximations (the Gatti and BH methods) alongside our novel approach.

## 3. Gatti Model: Decoupling Processes

### 3.1. Diffusion

Gatti et al. [46] analytically examined diffusion and repulsion as independent processes. In particular, note that we can extract the diffusion part from Equation (Equation 6) as
(7)$$\begin{array}{c} \partial_{t}Q^{\mathrm{diff}}=D\left(\partial_{RR}Q^{\mathrm{diff}}-\frac{2\partial_{R}Q^{\mathrm{diff}}}{R}\right) \end{array}$$
The associated diffusion equation for charge concentrations then becomes
(8)$$\partial_{t}n^{\mathrm{diff}}=D\nabla^{2}n^{\mathrm{diff}}$$
For an initial delta distribution, n(r,0)=δ(r), Equation (Equation 8) has a well-known solution:(9)$$n^{\mathrm{diff}}\left(r,t\right)=\frac{N}{{\left(4\pi Dt\right)}^{3/2}}\exp\left(-\frac{r^{2}}{4Dt}\right)$$
In other words, the charge concentration propagates following a 3D Gaussian shape with variance that progressively increases in time, as $\sigma {\left(t\right)}^{2}=2Dt$. If we consider instead an initial Gaussian distribution at the initial time, $n\left(r,0\right)∼N\left(0,\sigma_{0}\right)$, the same propagation shape is still valid. In that case, the concentration becomes
(10)$$n^{\mathrm{diff}}\left(r,t\right)=\frac{N}{{\left[2\pi \left(\sigma_{0}^{2}+2Dt\right)\right]}^{3/2}}\exp\left(-\frac{r^{2}}{2(\sigma_{0}^{2}+2Dt)}\right)$$
We can then calculate by integration [46] the solution of Equation (Equation 7), which is the initial diffusion equation for total charges.

### 3.2. Repulsion

On the other hand, the repulsion part is extracted from Equation (Equation 1) as
(11)$$\begin{array}{c} \partial_{t}Q^{\mathrm{rep}}=-\mu \frac{\partial_{R}Q^{\mathrm{rep}}}{4\pi R^{2}}\frac{Q^{\mathrm{rep}}}{\epsilon } \end{array}$$
Using the separation of variables method, Gatti et al. [46] found the following solution:(12)$$Q^{\mathrm{rep}}\left(R,t\right)=\frac{R^{3}}{3\left(\frac{\mu }{4\pi \epsilon }\right)t}\left[H\left(R\right)-H(R-R_{0}\left(t\right))\right]+qNH\left(R-R_{0}\left(t\right)\right)$$
where the time-dependent function $R_{0}\left(t\right)$ is defined as
(13)$$R_{0}\left(t\right)={\left[3\left(\frac{\mu }{4\pi \epsilon }\right)tqN\right]}^{1/3}$$
Note that beyond the radius $R_{0}\left(t\right)$, the charge distribution collects all the charges (qN coulombs). Here, we have used the Heaviside step function,
(14)$$H\left(x\right)=\left\{\begin{array}{cc} 0 & \mathrm{if}x<0,\\ 1 & \mathrm{if}x\geq 0. \end{array}\right.$$
Observe that in the limit when t→0, we obtain $R_{0}\left(t\right)=0$, and the total charge becomes $Q^{\mathrm{rep}}\left(R,0\right)=qN$ at R=0 and zero elsewhere. This corresponds to the delta distribution for the initial condition of the charge density, ρ(r,0)=qNδ(r). From Equation (Equation 12), we can calculate the charge concentration for t>0 using partial derivatives as
(15)$$\rho^{\mathrm{rep}}\left(r,t\right)=\left\{\begin{array}{cc} \frac{\epsilon }{\mu t} & \mathrm{if}r<R_{0}\left(t\right),\\ 0 & \mathrm{if}r\geq R_{0}\left(t\right). \end{array}\right.$$
In conclusion, at any given time, the charge density remains constant within the radius $R_{0}$ and zero beyond it. Thus, in contrast to the 3D Gaussian spread resulting from diffusion, repulsion cause the charges to propagate as a 3D uniform distribution.

An alternative approach to solving the repulsion-only equation can be found in terms of charge density [47], $\partial_{t}\rho^{\mathrm{rep}}=-\nabla \cdot \left(\mu \rho^{\mathrm{rep}}E\right)$, instead of using Equation (Equation 11) for total charge. From the numerical result shown in Figure 2h, it is evident that the charge density ρ(r,t) must be uniform, which we adopt as an ansatz. Thus, $\partial_{t}\rho^{\mathrm{rep}}=-\mu \left[E\cdot \nabla \rho^{\mathrm{rep}}+\rho^{\mathrm{rep}}\nabla \cdot E\right]=-\mu \rho^{\mathrm{rep}}\nabla \cdot E=-{\left(\rho^{\mathrm{rep}}\right)}^{2}\mu /\epsilon$. Here, we have used the Gauss’s law from Equation (Equation 2) and the identity $\nabla \rho^{\mathrm{rep}}=0$, which is derived from the ansatz. This allows us to solve the ODE directly, yielding the same solution as in Equation (Equation 15).

### 3.3. Root Mean Square Metric

Figure 2 shows the ground-truth numerical results for diffusion-only, repulsion-only, and diffusion–repulsion, which were found using the aforementioned IFDM approach. We used the common values for electrons in CZT crystals [20] at room temperature with μ=1000 cm/Vs and D=25 c$m^{2}$/s. There are four common alternatives to present the results, which are shown in the different rows of Figure 2:The number of elementary charges, N(R), within a given radial distance *R* from the center of the charge cloud, is displayed in Figure 2a–c. For *R* values larger than the charge cloud, N(R) will approximate the total number of generated electron–hole pairs (in this representative case, $P=2\times 10^{5}$ charges). Alternatively, this variable can be expressed as the total Coulomb charge, Q(R)=qN(R), or as the normalized charge distribution, $\hat{Q}\left(R\right)=Q\left(R\right)/qP$.The probability density function (PDF) of the charge distribution, $f\left(r,t\right)=\partial_{R}\hat{Q}\left(R,t\right)$, is shown in Figure 2d–f.The charge densities $\rho \left(r,t\right)=\partial_{R}Q\left(R\right)/4\pi R^{2}$, expressed in C/c$m^{3}$, are displayed in Figure 2g–i.The charge densities projected over the *x*-coordinate (due to symmetry, this projection is identical for all coordinates) become
(16)$$\begin{array}{c} \rho_{x}\left(x,t\right)=\int_{-\infty }^{\infty }dy\int_{-\infty }^{\infty }dz\rho \left(x,y,z,t\right)=\\ \int_{\left|x\right|}^{\infty }dr\int_{0}^{2\pi }d\phi \rho \left(r,\theta ,\phi ,t\right)r\sin\left(\theta \right)=\int_{\left|x\right|}^{\infty }\rho \left(r,t\right)2\pi rdr \end{array}$$
where one needs to consider the spherical symmetry of the density and θ=π/2.

We previously mentioned that the density $\rho^{\mathrm{diff}}\left(r,t\right)=q\;n^{\mathrm{diff}}\left(r,t\right)$ for the diffusion-only process is characterized by the shape of a 3D Gaussian, as derived by Gatti and presented in Equation (Equation 9). A fundamental principle in statistics [57] states that the marginal distribution of a multivariate normal distribution is also normally distributed. Thus, $\rho_{x}^{\mathrm{diff}}\left(x,t\right)$ is a 1D Gaussian (with the same mean and standard deviation), as observed in Figure 2j. This property will play an important role in the subsequent Benoit–Hamel approach.

Note that so far, our analysis has focused on the diffusion and repulsion of the charges distributed around a central point, the origin. To measure the spread of this distribution, we then consider the standard deviation of $\rho_{x}\left(x,t\right)$ around the zero mean, which provides the root mean squared:(17)$$\mathrm{RMS}\left(t\right)=\int_{\infty }^{\infty }x^{2}\rho_{x}\left(x,t\right)dx$$
Figure 3 indicates that the effect of repulsion on the cloud spreading is slightly stronger than that of diffusion. This behavior is influenced by the initial generation of electron–hole pairs, where diffusion dominates when the deposited energy is lower, while repulsion becomes the predominant factor as the deposited energy increases.

One can calculate the quadratic sum of the RMS for diffusion-only and repulsion-only to obtain a rough estimate of the RMS for the combined effects of diffusion and repulsion, as shown in Figure 3. Note that by decoupling both processes, we can provide an approximate efficient solution that no longer needs to solve PDEs, as we could directly use the analytical solutions.

## 4. BH Model: Gaussian Distribution

### 4.1. Analytical Derivations

Inspired by Gatti’s model, Benoit and Hamel (BH) [47] proposed an alternative approximate solution for the original system of Equations (Equation 1) and (Equation 2) but now effectively coupling both the diffusion and repulsion processes. They started with the initial assumption that the intricate projected charge density along the *x*-axis, resulting from the combined effects of diffusion and repulsion (refer to Figure 2l), can be approximated by a simple one-dimensional Gaussian density. Consequently, the following formula applies:(18)$$\rho_{x}^{\mathrm{BH}}\left(r,t\right)=\frac{Nq}{{\left[2\pi \sigma_{x}^{\mathrm{BH}}{\left(t\right)}^{2}\right]}^{1/2}}\exp\left(-\frac{r^{2}}{2\sigma_{x}^{\mathrm{BH}}{\left(t\right)}^{2}}\right)$$
where $\sigma_{x}^{\mathrm{BH}}=\sigma_{x}^{\mathrm{diff}}+\sigma_{x}^{\mathrm{rep}}$ is an unknown function responsible for the spread in time. Note that this assumption is equivalent to saying that the three-dimensional charge density ρ(x,y,z,t) can be approximated as a 3D multivariate normal density. We will now determine each of the two addends of the standard deviation:As mentioned above, charges diffusing in three dimensions adopt a Gaussian distribution with a variance of $\sigma^{\mathrm{diff}}{\left(t\right)}^{2}=2Dt$. The marginal density $\rho_{x}^{\mathrm{diff}}$ along the *x*-axis remains Gaussian and has the same identical variance, so $\sigma_{x}^{\mathrm{diff}}{\left(t\right)}^{2}=2Dt$. For reasons that will become apparent in Section 4.3, expressing this formula as a time-dependent differential equation is more advantageous. Thus, we could represent it as
(19)$$\partial_{t}\sigma_{x}^{\mathrm{diff}}{\left(t\right)}^{2}=2D$$On the other hand, the charges spread as a result of repulsion in a uniform three-dimensional shape. The charge density $\rho^{\mathrm{rep}}\left(r\right)$ due to repulsion-only is described in Equation (Equation 15), but note that the marginal density $\rho_{x}^{\mathrm{rep}}\left(x\right)$ along the *x*-axis has a more intricate mathematical description: As seen in Figure 2k, it is no longer uniform. Despite its complicated shape, the variance of this marginal density can still be calculated by definition. For instance, after a few calculations in the intermediate steps, one can derive the following:
(20)$$\sigma_{x}^{\mathrm{rep}}{\left(t\right)}^{2}=\int_{\infty }^{-\infty }{\left(x-0\right)}^{2}\rho_{x}^{\mathrm{rep}}\left(x,t\right)dx=\frac{R_{0}^{2}\left(t\right)}{5}$$As mentioned in the point above, we are now looking for an expression of the temporal derivative of the variance. Therefore, we easily derive the following expression:
(21)$$\partial_{t}\sigma_{x}^{\mathrm{rep}}{\left(t\right)}^{2}=\frac{2R_{0}^{\prime }\left(t\right)R_{0}\left(t\right)}{5}=\frac{\mu Nq}{10\sqrt{5}\pi \epsilon \sigma_{x}^{\mathrm{rep}}\left(t\right)}$$
Assuming that these two processes can be coupled together using a simple Gaussian distribution from Equation (Equation 18) with an evolving standard deviation $\sigma_{x}^{\mathrm{BH}}\left(t\right)$, one obtains
(22)$$\partial_{t}\sigma_{x}^{\mathrm{BH}}{\left(t\right)}^{2}=2\left(D+\frac{\mu Nq}{20\sqrt{5}\pi \epsilon \sigma_{x}^{\mathrm{BH}}\left(t\right)}\right)=2\tilde{D}\left(t\right)$$
In conclusion, the BH approach suggests that repulsion can also be modeled as a second-order process similar to diffusion. The model now includes a time-dependent coefficient, denoted by $\tilde{D}\left(t\right)$, which accounts for both diffusion and repulsion.

Note that another strength of the BH approach is that a similar mathematical procedure applies for elliptical (non-spherically symmetric) charge distributions. We can model these shapes using asymmetric 3D Gaussian distributions, where the diagonal covariance matrix may have $\sigma_{x}\neq \sigma_{y}\neq \sigma_{z}$. The asymmetry for initialization provides higher degrees of freedom and flexibility, which generally allows better fits to the experimental data).

### 4.2. Numerical Evaluation

The BH approach provides extremely accurate predictions for the spread of charges. The average absolute error for RMS is as low as 0.40 μm, while the RMS error found with the Gatti model (see Figure 3) was 4.20 μm. Figure 4a illustrates the comparison between the ground-truth distributions and those calculated using the BH approach. Although the BH model provides a good overall fit of the charge distributions, there is room for improvement: this method does not fully capture the exact shape of the charge cloud.

It is important to note that fitting the ground-truth numerical solutions with a simple Gaussian distribution is fully equivalent to using the Benoit–Hamel (BH) approach. In essence, the BH method inherently provides the optimal fit to the data. This property is evident because the Gaussian distribution involves only two parameters: a known mean of zero and a standard deviation, which is precisely derived from the ODE in Equation (Equation 22) to match the standard deviation of the data.

To show the discrepancy between the ground-truth and approximated normal distributions, we will measure the root mean squared error (RMSE) and the mean absolute error (MAE) at each time,
(23)$$\mathrm{RMSE}\left(t\right)=\sqrt{\frac{1}{N}\sum_{i=1}^{N}{\left[{\hat{Q}}^{\mathrm{gt}}\left(R_{i},t\right)-{\hat{Q}}^{\mathrm{approx}}\left(R_{i},t\right)\right]}^{2}}$$
(24)$$\mathrm{MAE}\left(t\right)=\frac{1}{N}\sum_{i=1}^{N}|{\hat{Q}}^{\mathrm{gt}}\left(R_{i},t\right)-{\hat{Q}}^{\mathrm{approx}}\left(R_{i},t\right)|$$

The time-averaged errors for the BH model were an MSE of 3.21% and an RMSE of 2.19%. Figure 4b shows the RMSE and MAE over time. Moreover, one can see how these errors tend to increase as the charges evolve over time and continue to spread.

### 4.3. Monte Carlo Algorithm

The advantage of using a simple Gaussian distribution approximation for both processes lies in the ability to simulate the diffusion–repulsion process as a *random walk*. Note that the marginal charge distribution along the *x*-axis, $\rho_{x}\left(x,t\right)$, follows $N\left[0,\sigma_{x}^{\mathrm{BH}}{\left(t\right)}^{2}\right]$ at any time step t∈{1,…,T=200}. To sequentially simulate this marginal distribution in a stochastic manner, we will now consider $X_{i}\left(t\right)$ random variables, for i∈{1,2,…,S} and S=200 for the number of samples, which are each distributed as
(25)$$X_{i}\left(t\right)∼X_{i}\left(t-1\right)+N\left[0,\Delta \sigma^{\mathrm{BH}}{\left(t\right)}^{2}\right]$$
The initial delta distribution at t=0 produced right after the photon absorption is often represented numerically as a Gaussian distribution with a small standard deviation. In our simulations, we initialize $X_{i}\left(0\right)∼N\left[0,\sigma^{\mathrm{BH}}{\left(0\right)}^{2}\right]$, where the standard deviation $\sigma^{\mathrm{BH}}\left(0\right)$ is 10 μm, as seen in [47]. Therefore, the progression iteratively expands to
(26)$$\begin{array}{c} X_{i}\left(t\right)∼N\left[0,\sigma^{\mathrm{BH}}{\left(0\right)}^{2}\right]+\\ \sum_{i=0}^{M}N\left[0,\sigma^{\mathrm{BH}}{\left(t-i\right)}^{2}-\sigma^{\mathrm{BH}}{\left(t-\left(i-1\right)\right)}^{2}\right] \end{array}$$
Note that the sum of independent Gaussian variables follows a Gaussian distribution [58] with the mean and variance being the sum of the individual means and variances. Applying this property to Equation (Equation 26), it can be seen that the random variables $X_{i}\left(t\right)$ will indeed follow $N\left[0,\sigma^{\mathrm{BH}}{\left(t\right)}^{2}\right]$, which is in agreement with the charge distribution described by the Benoit–Hamel method.

Observe that we initially had to solve the complicated non-linear PDE from Equation (Equation 6), which required the computationally intensive IFDM. In contrast, we can now iteratively propagate the *S* representative charges stochastically at each time step. Before each step, we need to find $\Delta \sigma^{\mathrm{BH}}{\left(t\right)}^{2}=\sigma^{\mathrm{BH}}{\left(t\right)}^{2}-\sigma^{\mathrm{BH}}{\left(t-1\right)}^{2}=2\tilde{D}\left(t\right)\Delta t$, as described in Equation (Equation 22). It becomes now clear why we reformulated the standard deviation using a time-dependent ODE. Moreover, we can then efficiently solve that equation using a first-order explicit finite-difference scheme.

Of course, the same approach followed for the propagation of charge in the *x*-direction can also be implemented for the *y* and *z* axes due to the spherical symmetry. The complete BH method is outlined in Algorithm 1, which takes only about 9–16 μs on a standard computer (hundreds of times faster than with the numerical IFDM). Note that, in the algorithm, the notation $N_{S}\left(\cdot ,\cdot \right)$ denotes the generation of *S* normally distributed samples with a certain mean and variance. The results are shown in Figure 5.

Additionally, it should be emphasized that the BH approach is compatible with existing detector simulators that account for additional charge displacements and fluctuations. At each time step, one can drift the *S* representative charges due to the bias E-field, then add a secondary weaker walk due to diffusion–repulsion, and finally recalculate the amount of charges after the trap-induced recombination process, as explained in more detail in [47].
**Algorithm 1** BH approach: Diffusion and repulsion**Require:** Simulation parameters  Number of samples, S=200  Time steps, T=200  Time increment, $\Delta t=5\times 10^{-9}$ s**Require:** Physical constants  Electron charge, $q=1.60\times 10^{-19}$ C  Permittivity, $\epsilon_{0}=8.85\times 10^{-12}$ F/m  Relative permittivity, $\epsilon_{r}=11$  Diffusion coefficient, $D=25\;{\mathrm{cm}}^{2}/s$  Electron mobility, $\mu =1000\;{\mathrm{cm}}^{2}/Vs$  Number of electrons, $P=2\times 10^{5}$  Initial standard deviation, $\sigma_{0}=10\;\mu$m  $\epsilon \leftarrow \epsilon_{0}\cdot \epsilon_{r}$  $\sigma_{\mathrm{BH}}^{2}\left[0\right]\leftarrow \sigma_{0}^{2}$  $x\left[0\right],y\left[0\right],z\left[0\right]∼N_{S}\left(0,1\right)\sigma_{0},N_{S}\left(0,1\right)\sigma_{0},N_{S}\left(0,1\right)\cdot \sigma_{0}$  **for **t←1 to *T*
**do**      $\tilde{D}\leftarrow D+\frac{\mu Pq}{20\sqrt{5}\pi \epsilon \sqrt{\sigma_{\mathrm{BH}}^{2}\left[t-1\right]}}$      $\Delta \sigma^{\mathrm{BH}}\leftarrow \sqrt{2\cdot \tilde{D}\cdot \Delta t}$      $x\left[t\right]\leftarrow x\left[t-1\right]+N_{S}\left(0,1\right)\cdot \Delta \sigma^{\mathrm{BH}}$      $y\left[t\right]\leftarrow y\left[t-1\right]+N_{S}\left(0,1\right)\cdot \Delta \sigma^{\mathrm{BH}}$      $z\left[t\right]\leftarrow z\left[t-1\right]+N_{S}\left(0,1\right)\cdot \Delta \sigma^{\mathrm{BH}}$      $\sigma^{\mathrm{BH}}{\left[t\right]}^{2}\leftarrow \Delta \sigma^{\mathrm{BH}}{\left[t-1\right]}^{2}+2\cdot \tilde{D}\cdot \Delta t$  **end for**

## 5. Our Proposed GND Model

### 5.1. Generalized Normal Distributions

Rather than approximating the normalized charge distribution $\hat{Q}\left(R,t\right)$ as a 3D Gaussian shape, we now propose that it more closely resembles a generalized normal distribution (GND). The probability density function (PDF) of the GND is defined as follows:(27)$$f^{\mathrm{GND}}\left(r,t\right)=\frac{\beta^{\mathrm{GND}}}{2\alpha^{\mathrm{GND}}\Gamma \left(1/\beta^{\mathrm{GND}}\right)}\exp{\left(\frac{|x-\mu^{\mathrm{GND}}|}{\alpha^{\mathrm{GND}}}\right)}^{\beta^{\mathrm{GND}}}$$
Note that we must consider the parameters as time-dependent, being for instance $\mu^{\mathrm{GND}}\left(t\right)$ the mean. Additionally, the mean is now not necessarily zero (see Figure 2d–f), since Equation (Equation 27) represents the radial density $f\left(r,t\right)=\partial_{R}\hat{Q}\left(R,t\right)$ and *not* the marginal density on the *x*-axis (Figure 2j–l). The parameter $\beta^{\mathrm{GND}}\left(t\right)$ modulates the peakedness of the distribution, categorizing a family of distributions. Specifically, $\beta^{\mathrm{GND}}=2$ corresponds to the standard normal distribution with mean $\mu^{\mathrm{GND}}\left(t\right)$ and variance $\alpha^{\mathrm{GND}}{\left(t\right)}^{2}/2$. Values of $\beta^{\mathrm{GND}}$ less than two result in distributions with sharper peaks and heavier tails, whereas values greater than two yield flatter densities with lighter tails.

Figure 4a presents the ground-truth numerical solutions alongside their fits using a Gaussian distribution (solid lines) and the GND (dashed lines, superimposed on the numerical solutions). As anticipated, the greater flexibility of the GND allows for a markedly improved fit to the numerical solutions. The RMS error decreases now even more than with the BH approach, reaching 0.18 μm with the GND. Similarly, Figure 4b illustrates the error between the ground-truth numerical solutions and their fits, comparing the simple Gaussian distribution (solid blue and red lines) with the GND (dashed blue and red lines). The corresponding time-average MAE and RMSE are exceptionally low for the GND fits with values of 0.17% and 0.14%, respectively. These errors are an order of magnitude lower than those obtained from simple Gaussian fits. More importantly, the error appears to be stable over time (see Figure 4b). We also evaluated additional metrics for comparison, such as the Kullback–Leibler (KL) divergence, the Pearson correlation coefficient, and the cosine similarity. The results, detailed in Table 1, further highlight the advantages of adopting the generalized normal distribution (GND) over the simple Gaussian distribution. Observe that for each time step, the parameters were found by solving a least squares error problem, which fits the ground-truth distribution to our proposed model using the Trust Region Reflective Optimizer defined in the Python package *SciPy* [59].

### 5.2. Gaussian Mixture Model

We aim to develop a Monte Carlo simulation based on Gaussian processes, as previously shown in Section 4.3. This will enable the integration of our approximated results for repulsion–diffusion into a broader simulator that also accounts for the additional effects of charge drift and fluctuations. To build the stochastic method, we will first apply a Gaussian Mixture Model (GMM) to model the generalized normal distribution (GND) as a sum of Gaussian distributions. Our findings indicate that the sum of only two Gaussian components is sufficient to accurately represent the GNDs that appear in our problem. The PDF of the GMM is then expressed as follows:(28)$$f^{\mathrm{GMM}}\left(r\right)=w_{1}\phi \left(r;\mu_{1},\sigma_{1}^{2}\right)+w_{2}\phi \left(r;\mu_{2},\sigma_{2}^{2}\right)$$

Here, ϕ represents the PDF of a Gaussian distribution, and the parameters $\left(\mu_{1},\sigma_{1}^{2}\right)$ and $\left(\mu_{2},\sigma_{2}^{2}\right)$ specify the means and variances of the two Gaussian components, respectively. Note that the GND is symmetric around its central point $\mu_{\mathrm{GND}}$. We will maintain such a property by ensuring that both Gaussian functions have (1) the same weight $w_{1}=w_{2}=0.5$, (2) the same standard deviation $\sigma_{1}=\sigma_{2}$, and (3) symmetric means with $|\mu^{\mathrm{GND}}-\mu_{1}\left|=\right|\mu^{\mathrm{GND}}-\mu_{2}|$. Let us now determine the exact values for these four unknown parameters:We can find a reasonable value for $\mu_{1}$ (and therefore $\mu_{2}$ also) by matching the first three moments (mean, variance, and skewness) of the GMM with the GND. In particular, if we define
(29)$$\mu_{1}\left(t\right):=\sqrt{\sigma^{\mathrm{GND}}{\left(t\right)}^{2}-\sigma_{1}{\left(t\right)}^{2}}+\mu^{\mathrm{GND}}$$Then, we can derive, from Equation (Equation 29), the following
(30)$$\mu^{\mathrm{GMM}}=w_{1}\mu_{1}+w_{2}\mu_{2}=\mu^{\mathrm{GND}}$$
(31)$${\left(\sigma^{\mathrm{GMM}}\right)}^{2}=w_{1}\sigma_{1}^{2}+w_{2}\sigma_{2}^{2}+w_{1}{\left(\mu_{1}-\mu^{\mathrm{GMM}}\right)}^{2}+w_{2}{\left(\mu_{2}-\mu^{\mathrm{GMM}}\right)}^{2}={\left(\sigma^{\mathrm{GND}}\right)}^{2}$$Note that both results are valid for any $\sigma_{1}$ value (we will decide on a particular value below). Because both the GMM and the GND are symmetric, there is also a match of the third moment (the skewness is zero).We can now find a convenient $\sigma_{1}$ value by matching the peak of the GMM density with the peak of the GND density at the central point, $r_{0}=\mu^{\mathrm{GND}}$, meaning
(32)$$f^{\mathrm{GND}}\left(r_{0};\mu^{\mathrm{GND}},\sigma^{\mathrm{GND}}\right)\equiv f^{\mathrm{GMM}}\left(r_{0};\mu_{1},\sigma_{1}\right)$$Substituting the value of $\mu_{1}$ from Equation (Equation 29) and considering the known peak value $c_{\mathrm{peak}}$ of the GND distribution, one finds
(33)$$f^{\mathrm{GMM}}\left(\sigma_{1}\right)=\frac{1}{\sqrt{2\pi \sigma_{1}^{2}}}\exp\left(\frac{{\left(\sigma^{\mathrm{GND}}\right)}^{2}-\sigma_{1}^{2}}{2\sigma_{1}^{2}}\right)=c_{\mathrm{peak}}$$This non-linear algebraic equation can be efficiently solved as a least squares minimization problem, where the objective function represents the residuals of the equation. Specifically, we employed the Python package *SciPy* and utilized again the efficient trust-region optimizer, incorporating box constraints to ensure that $\sigma_{1}$ remains within the physically meaningful range of positive values less than the known parameter $\sigma^{\mathrm{GND}}$.

Figure 6 shows the fits of the GND probability density functions at two different times using the proposed mixture model.

### 5.3. Monte Carlo Algorithm

Using Equations (Equation 29) and (Equation 32), we can also compute the values for $\left(\mu_{1},\sigma_{1}\right)$ and $\left(\mu_{2},\sigma_{2}\right)$. We now have all the necessary information to develop a Monte Carlo simulator with the proposed GND approach. This simulator follows a methodology analogous to the one described in Algorithm 1 with the slight difference of (1) having two simultaneous Gaussian processes instead of one, and (2) doing the updates of the radial distance instead of the Cartesian coordinates. The new methodology is outlined in Algorithm 2. The computational cost of this modified algorithm ranges between 16 and 19 μs on a standard computer, which is approximately double that of the original BH approach. The charge distributions found with this simulation are illustrated in Figure 5.
**Algorithm 2** Novel approach: Diffusion and repulsion**Require:** Simulation parameters (as in Algorithm 1)**Require:** Physical constants (as in Algorithm 1)**Require:** Pre-computed parameters: $\mu_{1},\mu_{2},\sigma_{1},\sigma_{2}$  $\epsilon \leftarrow \epsilon_{0}\cdot \epsilon_{r}$  $r_{1}\left[0\right]\leftarrow \mu_{1}\left[0\right]+N_{S}\left(0,\sigma_{1}\left[0\right]\right)$  $r_{2}\left[0\right]\leftarrow \mu_{2}\left[0\right]+N_{S}\left(0,\sigma_{2}\left[0\right]\right)$  $\phi_{1},\phi_{2}\leftarrow {\mathrm{rand}}_{S}\left[0,2\pi \right]$  $\theta_{1},\theta_{2}\leftarrow {\mathrm{rand}}_{S}\left[0,\pi \right]$  $x_{1}\left[0\right],x_{2}\left[0\right]\leftarrow r_{1}\left[0\right]\sin\phi_{1}\cos\theta_{1},r_{2}\left[0\right]\sin\phi_{2}\cos\theta_{2}$  $y_{1}\left[0\right],y_{2}\left[0\right]\leftarrow r_{1}\left[0\right]\sin\phi_{1}\sin\theta_{1},r_{2}\left[0\right]\sin\phi_{2}\sin\theta_{2}$  $z_{1}\left[0\right],z_{2}\left[0\right]\leftarrow r_{1}\left[0\right]\cos\phi_{1},r_{2}\left[0\right]\cos\phi_{2}$  **for **t←1 to *T*
**do**      $\Delta \mu_{1}\left[t\right]\leftarrow \mu_{1}\left[t\right]-\mu_{1}\left[t-1\right]$      $\Delta \mu_{2}\left[t\right]\leftarrow \mu_{2}\left[t\right]-\mu_{2}\left[t-1\right]$      $\Delta \sigma_{1}\left[t\right]\leftarrow \sigma_{1}\left[t\right]-\sigma_{1}\left[t-1\right]$      $\Delta \sigma_{2}\left[t\right]\leftarrow \sigma_{2}\left[t\right]-\sigma_{2}\left[t-1\right]$      $\Delta r_{1}\left[t\right]\leftarrow \Delta \mu_{1}\left[t-1\right]+N_{S}\left(0,1\right)\cdot \Delta \sigma_{1}$      $\Delta r_{2}\left[t\right]\leftarrow \Delta \mu_{2}\left[t-1\right]+N_{S}\left(0,1\right)\cdot \Delta \sigma_{2}$      $x_{1}\left[t\right]\leftarrow x_{1}\left[t-1\right]\left(1+\Delta r_{1}/r_{1}\left[t-1\right]\right)$      $x_{2}\left[t\right]\leftarrow x_{2}\left[t-1\right]\left(1+\Delta r_{2}/r_{2}\left[t-1\right]\right)$      $y_{1}\left[t\right]\leftarrow y_{1}\left[t-1\right]\left(1+\Delta r_{1}/r_{1}\left[t-1\right]\right)$      $y_{2}\left[t\right]\leftarrow y_{2}\left[t-1\right]\left(1+\Delta r_{2}/r_{2}\left[t-1\right]\right)$      $z_{1}\left[t\right]\leftarrow z_{1}\left[t-1\right]\left(1+\Delta r_{1}/r_{1}\left[t-1\right]\right)$      $z_{2}\left[t\right]\leftarrow z_{2}\left[t-1\right]\left(1+\Delta r_{2}/r_{2}\left[t-1\right]\right)$      $r_{1}\left[t\right]\leftarrow \sqrt{x_{1}{\left[t\right]}^{2}+y_{1}{\left[t\right]}^{2}+z_{1}{\left[t\right]}^{2}}$      $r_{2}\left[t\right]\leftarrow \sqrt{x_{2}{\left[t\right]}^{2}+y_{2}{\left[t\right]}^{2}+z_{2}{\left[t\right]}^{2}}$**  end for**

### 5.4. Extending the Algorithm for Different Scenarios

One can observe that there is a high correlation between the parameter $\sigma^{\mathrm{BH}}\left(t\right)$ found with the simple Gaussian fit and the parameters $\mu^{\mathrm{GND}}\left(t\right)$ and $\alpha^{\mathrm{GND}}\left(t\right)$ of the GND approach. This correlation is clearly illustrated in Figure 7a,b. The consistency between the two models allows us to generalize our proposed GND for other scenarios (with different material properties and deposited energies) without having to pre-compute the GND parameters for each scenario beforehand rather using the very convenient results from the Benoit–Hamel model.

For a specific material with permittivity ϵ, the dynamic of the charges due to diffusion and repulsion dynamics are solely influenced by the charge mobility μ and the number of electron–hole pairs-generated *P*. Indeed, it is important to remark again that the diffusion coefficient *D* is not an independent variable; as it is linearly related to μ by the Einstein–Smoluchowski relation [53]. Taking for example a wide range of interest, such as μ∈[700,1300] for electrons in CZT and $P\in [2\times 10^{4},2\times 10^{5}]$, we have established the following relationships:(34)$$\mu^{\mathrm{GND}}\left(t;\mu ,P\right)=\left[\mu a_{1}+Pb_{1}+c_{1}\right]\cdot \sigma^{\mathrm{BH}}\left(t\right)+\left[\mu d_{1}+Pe_{1}+f_{1}\right]$$
(35)$$\alpha^{\mathrm{GND}}\left(t;\mu ,P\right)=\left[\mu a_{2}+Pb_{2}+c_{2}\right]\cdot \sigma^{\mathrm{BH}}\left(t\right)+\left[\mu d_{2}+Pe_{2}+f_{2}\right]$$

The particular values of the coefficients are provided in Appendix B for these particular ranges of interest. Similar analyses could be conducted by the reader in other different ranges of interest for μ and *P* to analyze different crystals.

It is important to note that shaping parameter $\beta^{\mathrm{GND}}$ shows a minimal dependency on the physical parameters (μ,P), and therefore a correlation with $\sigma^{\mathrm{BH}}$ was not found. This time-dependent coefficient influences the peakedness of the distribution, and a value greater than two allows for the steeper distributions observed in our problem (see Figure 4a). Our findings indicate that the time evolution of $\beta^{\mathrm{GND}}\left(t\right)$ aligns closely with a model that features an initial exponential decay followed by a gradual transition to a linear decrease. We have modeled this parameter as a piecewise function with $\beta^{\mathrm{GND}}\left(t=0\right)=a_{3}$, matching the initial distribution at the starting time, and then modeling the time evolution for t>0 as
(36)$$\beta^{\mathrm{GND}}\left(t\right)=b_{3}e^{-c_{3}t}+\left(d_{3}t+e_{3}\right)$$
The particular coefficients are also provided in the Appendix B. The R-squared of the $\beta^{\mathrm{GND}}$ parameter from Equation (Equation 36), calculated with respect to different ground-truth $\beta^{\mathrm{GND}}\left(t;\mu ,P\right)$, is on average 0.986, showing an excellent fit, as one can observe in Figure 7c.

Employing the time-dependent parameters from Equations (Equation 35) to (Equation 36) using the presented correlation analysis, we obtained the time-averaged MAE of 0.44% and RMSE of 0.35% for the charge cloud distribution. These results are slightly worse than those obtained from pre-computed parameters (shown in Table 1), but they are still about an order of magnitude better than the results from the BH approach. In exchange for losing some accuracy, this correlation-based method offers greater flexibility: It extends our novel algorithm to different scenarios without the need to pre-compute any values at all, just using the simple solutions for $\sigma^{\mathrm{BH}}\left(t\right)$ from the Benoit–Hamel algorithm.

## 6. Concluding Remarks

In this study, we examine the charge dynamics in semiconductors, focusing on diffusion and repulsion effects, with specific applications to high-energy semiconductor detectors.

First, we treated diffusion and repulsion as decoupled processes and used the theoretical findings from Gatti et al. [46] to obtain analytical expressions for the charge distributions. While the charges propagate following a multivariate (3D) Gaussian distribution under the influence of diffusion, they propagate as a multivariate (3D) uniform distribution due to Coulomb repulsion. Considering the RMS as the quadratic sum of both separate processes leads to considerable inaccuracies. In particular, in our simulations for electron dynamics in a CZT detector, the time-average RMS mismatch for the charge cloud distribution was found to be 4.20 μm. On the bright side, these approximated results can be computed very efficiently from the close-form expressions.

Second, we analyzed how the Benoit–Hamel (BH) approach [47] improves upon the previous model by coupling both processes, approximating the overall charge distribution as a simple multivariate 3D Gaussian function. This methodology enables an efficient implementation of the distributions using a Monte Carlo technique. It also allows for the easy integration of these two weak processes in more general simulation frameworks that account for other effects. Note that the standard deviation of the Gaussian distribution increases with time, reflecting the spatial expansion of the charges. In our simulations, the BH approach yielded a time-averaged RMS error of only 0.40 μm. Although this method provides high accuracy in predicting the spread of charges, the actual shape of the charge distribution does not closely resemble a Gaussian function. For instance, the MAE and RMSE of the charge distribution were 3.21% and 2.19%, respectively.

Third, we have developed a novel method that naturally extends the BH approach. Rather than taking a simple Gaussian as an ansatz, we consider a generalized normal distribution (GND). This new assumption provides higher degrees of freedom that allow better fits of the actual ground-truth distributions. We approximated the GND as a Gaussian Mixture Model (GMM) with two Gaussian functions, enabling an efficient simulation of these processes using another Monte Carlo algorithm. Fitting the GND to the ground-truth numerical solutions lead to an RMS error of the cloud of 0.18%. The MAE and RMSE of the charge distribution (cloud shape) are also reduced to 0.17% and 0.14%. These results then demonstrate the strong capability of this more complex distribution to accurately model the system. The GND algorithm can be easily extended to various scenarios (with different material properties and deposited energies) by determining the GND parameters from the BH parameters via correlation analysis. Using this correlation approach, the error between the ground-truth numerical solution and the GND results remains low with an MAE of 0.44% and an RMSE of 0.35% for the cloud shape. The computational cost of the GND approach is about double that of the BH approach, although it is still reasonable and in the order of tens of milliseconds in standard computers.

These findings facilitate the refinement of semiconductor detector simulators and contribute to the analysis of next-generation detectors. Using these simulators, future research could focus on accurately solving inverse problems considering diffusion and repulsion either by performing event reconstructions or by characterizing the material properties of the detector bulk.

## Figures and Tables

**Figure 1 sensors-24-07123-f001:**
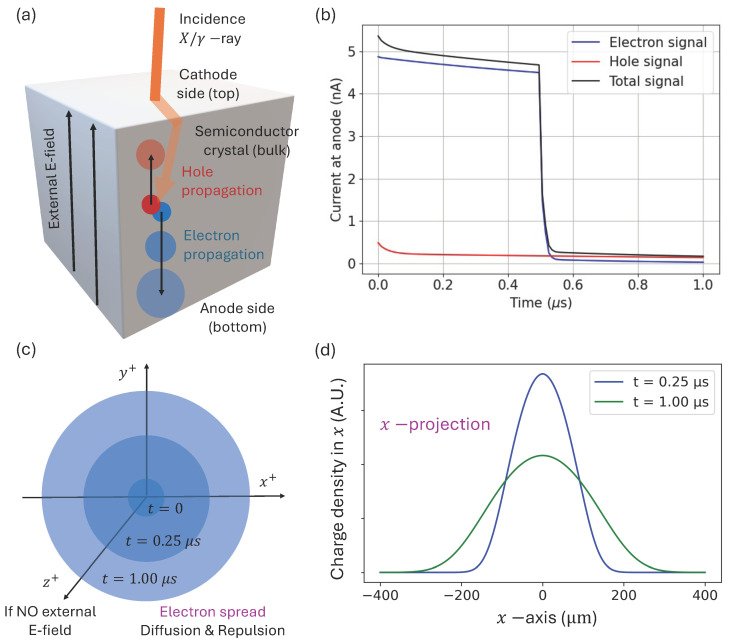
(**a**) Basic planar detector configuration. The absorption of incidence photons creates electron–hole pairs, and the free charges move through the bulk of the detector drift by the external E-field. (**b**) Simulated signals at the anode. We have used the specifications from [20] for CZT detectors. The simulator includes the charge drift due to the external field and the charge fluctuations (recombination, trapping and detrapping). (**c**) Spread of electrons in time due to diffusion and repulsion. If we compensate for the transport due to external E-field, the spread appears spherically symmetric around a central point. (**d**) Projected charge density on the *x*-direction at two different times.

**Figure 2 sensors-24-07123-f002:**
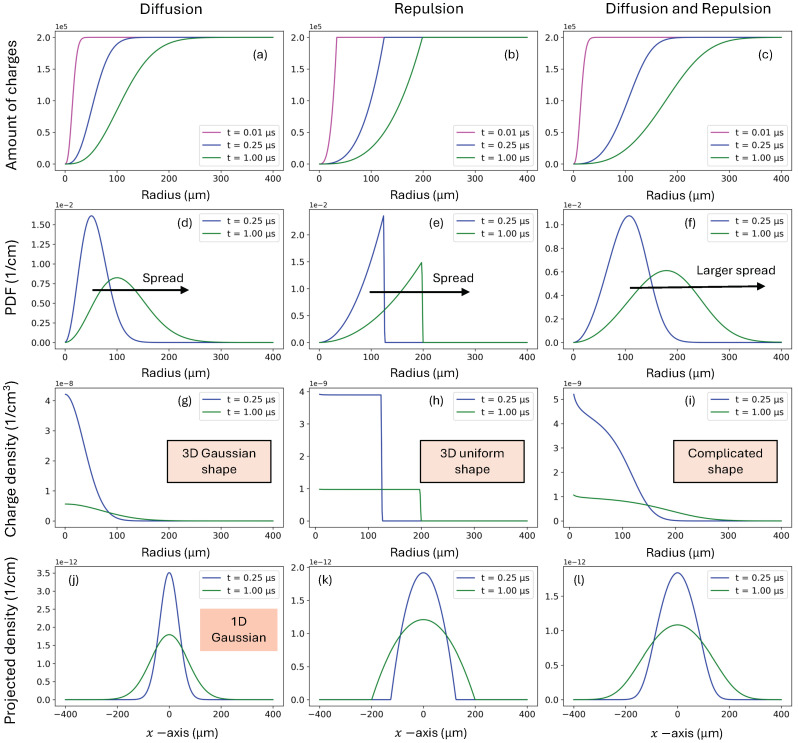
Numerical solutions from the IFDM for diffusion-only (left column), repulsion-only (middle column), and diffusion–repulsion combined (right column). The rows provide different (equivalent) expressions for the solutions, including the amoung of charges, the p Further description provided in Section 3.3.

**Figure 3 sensors-24-07123-f003:**
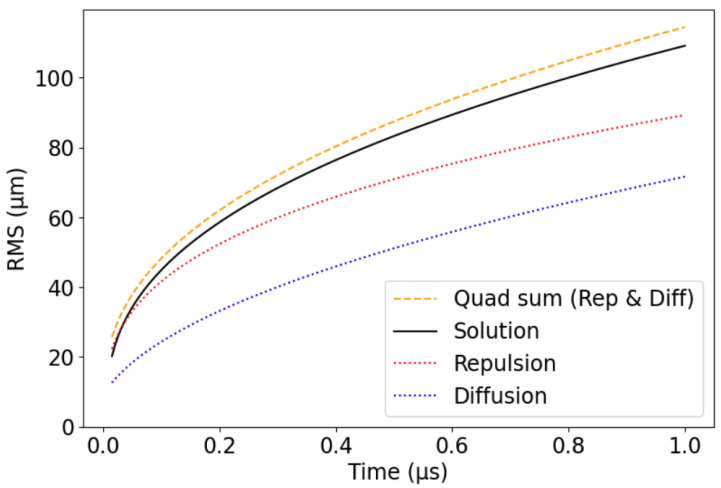
Root mean square for diffusion-only, repulsion-only, and diffusion–repulsion combined. We can see that the quadratic sum of diffusion-only and repulsion-only differs from the actual spread of the diffusion–repulsion processes. Parameters: μ=1000 cm/Vs, D=25 c$m^{2}$/s, $P=2\times 10^{5}$ electrons, $\epsilon_{r}=11$ relative permittivity, initial standard deviation $\sigma_{0}=10$μm.

**Figure 4 sensors-24-07123-f004:**
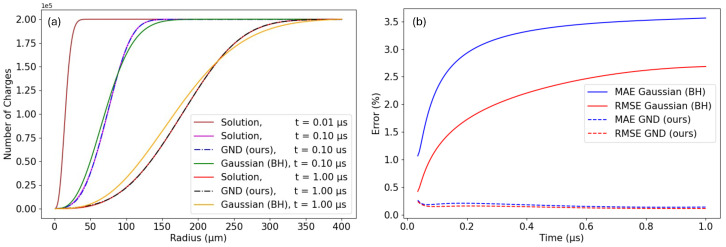
(**a**) Charge distribution (non-normalized) accounting for the total amount of elementary charges plotted at different times. We see the ground-truth distributions and those generated with the Benoit–Hamel model and with our proposed generalized normal distribution (GND) model. (**b**) Error metrics of the BH and our GND models.

**Figure 5 sensors-24-07123-f005:**
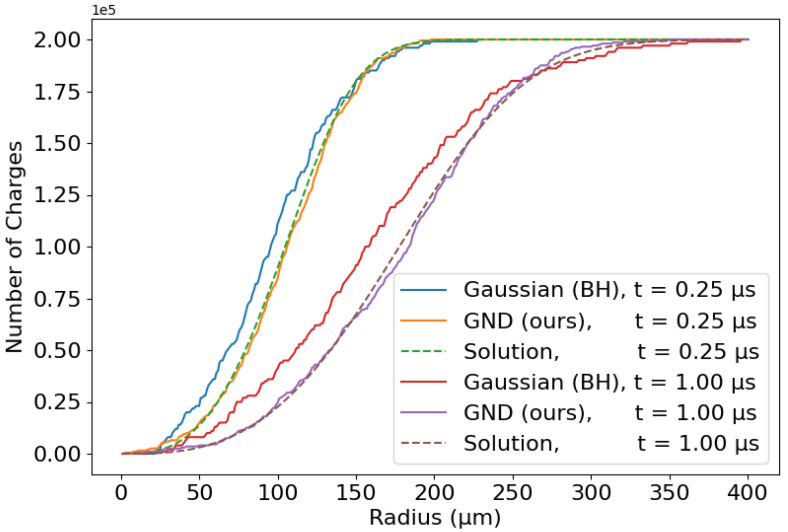
Monte Carlo simulation with S=200 representative samples, using the simple Gaussian model (BH) and the GND model (ours).

**Figure 6 sensors-24-07123-f006:**
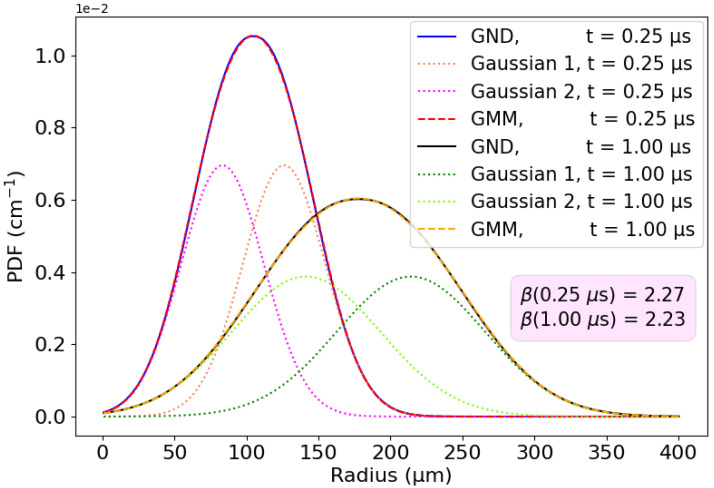
Gaussian Mixture Model (with two Gaussian functions) matching the PDF of the charges at two different times.

**Figure 7 sensors-24-07123-f007:**
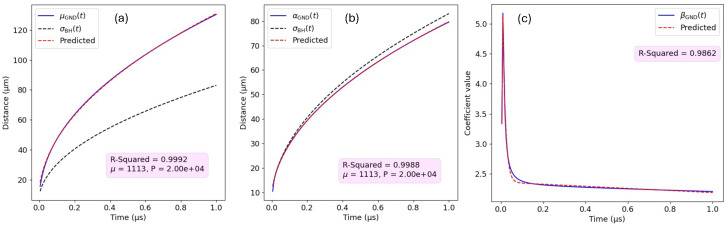
Illustrating the strong linear correlation between the time-dependent parameters (**a**) $\mu^{\mathrm{GND}}$ and $\sigma^{\mathrm{BH}}$, (**b**) $\alpha^{\mathrm{GND}}$ and $\sigma^{\mathrm{BH}}$, for a representative scenario with $P=2\times 10^{4}\in \left[2\times 10^{4},2\times 10^{5}\right]$ electron–hole pairs and with an electron mobility of μ=1113∈[700,1300] c$m^{2}$/Vs. (**c**) Time-average $\beta^{\mathrm{GND}}$ for different (P,μ) scenarios compared with our proposed model from Equation (Equation 36).

**Table 1 sensors-24-07123-t001:** Comparison of metrics: fitting ground-truth distributions using the simple Gaussian model by BH and our proposed GND method.

Metrics	Gaussian (BH)	GND (Ours)
RMS absolute error (μm)	0.40	0.18
MAE (%)	3.21	0.17
RMSE (%)	2.19	0.14
Cosine similarity	1–$2.19\cdot 10^{-3}$	1–$8.92\cdot 10^{-6}$
Correlation coefficient	1–$9.20\cdot 10^{-4}$	1–$2.14\cdot 10^{-6}$
KL divergence	$1.94\cdot 10^{-4}$	$1.32\cdot 10^{-5}$

## Data Availability

Data underlying the results presented in this paper are not publicly available at this time but may be obtained from the authors upon reasonable request.

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
