# Peer review of "Charge Diffusion and Repulsion in Semiconductor Detectors"

_sensors, 2024, doi:10.3390/s24227123_

Round 1
Reviewer 1 Report
Comments and Suggestions for Authors
This paper represents important progress for the simulation of charge propagation through semiconductor devices. The results will be beneficial to two applications: Simulation of novel devices to optimise design prior to production; Simulation of existing devices to better understand signal formation and reconstruction of observables.
The paper focusses of the description of the time-dependent expansion of the charge cloud from primary ionisation by diffusion and Coulomb repulsion. It starts out with a critical review of existing approaches to these coupled effects. The Gatti method decouples the two into a Gaussian expansion due to diffusion and a uniform expansion due to repulsion. The Benoit-Hamel method describes their coupled effect by a single multivariate Gaussian spread, with a time-dependent width. Both are carefully compared to the numerical solution of the coupled PDE, which describe the true physics of the processes. It is found that the Gatti method presents the advantage of a closed solution at the price of important inaccuracies in describing the charge cloud. The Benoit-Hamel method scores better, but describes only the width of the cloud (with several percent errors) and not its detailed shape.
Based on these findings the authors propose improvements to the Benoit.-Hamel approach, by generalising the single multivariate Gaussian to a Generalised Normal Distribution. Not only does such a function much better represent the cloud shape, it also reduces the errors for its parameters by more than an order of magnitude. In addition the authors propose approximations to their method, which ease the implementation in a Monte-Carlo simulation without spoiling the accuracy. The end result is a very high fidelity representation of the true charge distribution as a function of time, with tolerable computational cost.
The paper contains concrete algorithms and methods for parameter determination, such that an implementation of the improved methodology into existing simulators should be straight forward.
The following are a few formal remarks to improve the readability of the paper:
Lines 155-156: units should be µm and µs
Footnote 3, p. 6: Eq.??
Caption of Fig. 2: Section 3.3
Fig. 3: Colors for repulsion and diffusion curves are indistinguishable
Line 250: Section 4.3
Equ. 20: Integral from -infty to +infty
Footnote 4, p. 9: covariance matrix
Line 300: solved
Footnote 5, p. 11: script N
Line 430: Section 4.3
Line 444: ansatz
Author Response
Thank you, please find the responses in the attached file.

Reviewer 2 Report
Comments and Suggestions for Authors
Summary:
This paper deals with the modelization of the diffusion and repulsion phenomena taking place during the charge carriers' migration towards the electrodes in a biased semiconductor detector, in view of efficiently incorporating these phenomena in Monte Carlo simulations of the signal evolution for such kind of detectors.
These phenomena are often neglected in the simulations due to the high computational cost compared to the relatively minor contribution they have in the overall evolution of the system.
This work starts by evaluating two models already available in the literature (Gatti and Benoit-Hamel) and then proposes a novel treatment based on a generalized normal distribution modeled by means of a gaussian mixture. Results are compared to those obtained with the (computationally expensive) numerical solution of the diffusion-repulsion equation.
General comment:
The paper is globally well-written and presents interesting developments: I just have one major concern regarding how the comparison between the models is carried out, that I describe in the following.
The first part of the paper has an almost pedagogical tone, and the first two models are described quite in detail: there, using just the reference may have been enough, helping to cut long portions of the text.
Nevertheless, I have to point out that such introductory part is very well-written and can really help a general reader to picture the framework of this work. Therefore, despite the additional length, I would advise to keep it as it is.
Overall, the article is worthy of publications, after the following major revisions are addressed by the authors.
Major revisions:
l.60 Is the simulated Ebias assumed constant along the depleted region? I would not expect that in a reversely biased diode close to depletion operation.
eq.12 Please check this equation. The behavior obtained from that is not physically possible (for R<R0 the charge is >qN?). It would make sense if the last step function used was H(R-R0(t)).
l.334-337 and fig.4 (general remark): If I understand correctly, in Fig. 4 and throughout this subsection, the comparison is made between: (a) the numerical solution of eq.6 (i.e., the ground-truth solution), (b) the solution obtained using the BH method (which is still based on physical constants), and (c) the result obtained by modeling Q(R,t) as a Gaussian Normal Distribution, as proposed by the authors, with the parameters of the distribution fitted to the solution from (a).
If this interpretation is correct, I believe that comparing results (b) and (c) to (a) may not be entirely appropriate. Since (c) is derived by fitting to (a), it is trivially more similar to (a). While it is perfectly valid to present the result of the fitting procedure for (c), presenting it as a comparison alongside (b) could be misleading, as it may give the impression that (c) is an independent result rather than a fit to (a).
Could the authors clarify or comment on this point?
l.412 This comment is related to the one above. Here the parameters for method (c) are set based on the result of (b) via this correlation analysis, therefore the main difference between (b) and (c) lies indeed in the assumption of the shape of the charge distribution. If the MAE and RMSE reported at line 412 are still computed w.r.t. the ground-truth solution (a) (and not between (b) and (c) - am I interpreting correctly?), I believe that this is the actual result that should be emphasized, since with the GND one clearly obtains a better match with the numerical solution w.r.t. the values reported at line 276 for BH.
l.449 In line with the previous comment, the conclusions should emphasize the results from sec.5.4, as they reflect a more significant comparison. I would like the authors to comment on this point, especially if I have misunderstood any aspects of the interpretation.
Minor revisions:
l.44 In the sentence "When a photon is absorbed, it generates P electron-hole pairs [...]" it could be mentioned that between the two phenomena, there is first an actual mechanism of conversion of the photon before it is able to produce e-h pairs. It is not necessary, but would be nice.
l.87 Among the factors that impact the studied phenomenon (particularly diffusion), the operational conditions (e.g. temperature) should be listed too.
l.~120 For clarity, I suggest to specify also that E in that formula is the electric fiels generated by the cloud itself.
l.130 The first time it appears, it is preferable to spell out 'partial differential equation' (PDE), after which the acronym can be used. This applies to all acronyms in the manuscript.
l.228,229 In this sentence, 'energy levels' is mentioned twice. As I interpret it, the authors are referring to the amount of energy deposited in the active area of the detector, which is proportional to the number of e-h pairs generated. However, in physics, the term 'energy levels' is typically used to describe discrete energy states of quantum systems. I suggest using a different phrasing to avoid confusion.
Typos:
l.105 centered -> center
l.116 at A electron drift velocity -> at THE electron drift velocity
l.155,156 There are two \mu symbols which are not correctly displayed.
footnote n.3 The equation reference did not work. It should be Eq.2.
l.202 Fig.1 -> Fig.2
l.300 solved -> solve
l.340 Section 4.C -> 4.3
l.444 ansat -> ansatz
Author Response
Please find the responses in the file attach.
Thank you!

Round 2
Reviewer 2 Report
Comments and Suggestions for Authors
I thank the authors for the comments and clarifications they sent in response to my comments.
The modifications done have satisfactorily improved the clarity of the paper, which in my opinion can be accepted in the current form.